

# Land-cover change in Cuba and implications for the area of distribution of a specialist's host-plant

Claudia Nuñez-Penichet[1], Juan Maita[2] and Jorge Soberon[1]

[1] Biodiversity Institute and Department of Ecology & Evolutionary Biology, University of Kansas, Lawrence, Kansas, United States
[2] Carrera de Ingeniería Forestal, Centro de Investigaciones Tropicales del Ambiente y Biodiversidad, Universidad Nacional de Loja, Loja, Ecuador

## ABSTRACT

Changes in land cover directly affect biodiversity. Here, we assessed land-cover change in Cuba in the past 35 years and analyzed how this change may affect the distribution of *Omphalea* plants and *Urania boisduvalii* moths. We analyzed the vegetation cover of the Cuban archipelago for 1985 and 2020. We used Google Earth Engine to classify two satellite image compositions into seven cover types: forest and shrubs, mangrove, soil without vegetation cover, wetlands, pine forest, agriculture, and water bodies. We considered four different areas for quantifications of land-cover change: (1) Cuban archipelago, (2) protected areas, (3) areas of potential distribution of *Omphalea*, and (4) areas of potential distribution of the plant within the protected areas. We found that "forest and shrubs", which is cover type in which *Omphalea* populations have been reported, has increased significantly in Cuba in the past 35 years, and that most of the gained forest and shrub areas were agricultural land in the past. This same pattern was observed in the areas of potential distribution of *Omphalea*; whereas almost all cover types were mostly stable inside the protected areas. The transformation of agricultural areas into forest and shrubs could represent an interesting opportunity for biodiversity conservation in Cuba. Other detailed studies about biodiversity composition in areas of forest and shrubs gain would greatly benefit our understanding of the value of such areas for conservation.

## INTRODUCTION

Habitat degradation is one of the most important anthropogenic causes of biodiversity loss, driven mostly by changes in land use (*Leemans & de Groot, 2003*; *Baude, Meyer & Schindewolf, 2019*; *Santos et al., 2020*; *Stanturf, 2021*). Habitat loss often involves deteriorating habitat quality and habitat fragmentation which have ecological, genetic, and evolutionary consequences (*Sih, Jonsson & Luikart, 2000*; *Hanski, 2011*). This is a problem for most species, but it may be especially important for those with a restricted distribution or those that depend on a specific climate or habitat for survival (*Devictor, Julliard & Jiguet, 2008*). How habitat modifications are affecting species has been the focus of

Corresponding author
Claudia Nuñez-Penichet,
claudianunez@ku.edu

attention in several studies (*Brooks et al., 2002*). Those studies have been mainly focused on plants, birds, and mammals (*Zechmeister et al., 2003*; *de Lima et al., 2013*; *Powers & Jetz, 2019*). However, other groups like insects are underrepresented (*Basset & Lamarre, 2019*), despite documented examples of extinctions due to habitat loss (*Dunn, 2005*).

Land cover and land use changes have a large effect in the current distribution of insects' species (*Bommarco et al., 2014*). Some species are highly sensitive to these changes, experiencing population decline in areas where the non-natural cover types increase (*Fox et al., 2014*). For example, some studies have reported that bee population has been declining due to habitat cover change (*Nemésio et al., 2016*). The effects of land use change on species richness/diversity on Lepidoptera, on the other hand, do not have a consensus among previous studies as other factors, like habitat heterogeneity, are also important (*Nuñez-Penichet et al., 2021*). In this group, some studies reported that the butterfly communities are more diverse in undisturbed (or the least disturbed) forest than in disturbed habitats, and others reported the opposite trend (*Koh, 2007*).

The neotropical genus *Urania* (Uraniidae) includes four species of diurnal moths, *U. sloanus* (endemic to Jamaica, presumed extinct; *Lees & Smith, 1991*), *U. fulgens* (from Mexico to northern Colombia), *U. leilus* (from southern Colombia to Bolivia), and *U. boisduvalii* (endemic to Cuba; *Nazari et al., 2016*). These moths feed, during their larvae stages, exclusively on *Omphalea* (*Euphorbiaceae*) plants (*Lees & Smith, 1991*; *Smith, 1991*, *1992*; *Nuñez-Penichet & Barro, 2020*). These plants have toxins that protect them from insects, but *Urania* larvae can tolerate them (*Smith, 1992*). However, when *Omphalea* plants are being eaten for several generations of these moths, they increase the levels of toxicity as a defense mechanism against *Urania* herbivory (*Lees & Smith, 1991*), forcing them to move to a different host-plant population (*Smith, 1983*). Therefore, *Urania* needs to have several patches of *Omphalea* available to guarantee its survival.

In Cuba, there are three species of *Omphalea* plants (*Lees & Smith, 1991*) mainly distributed in coastal and other karstic zones with primary or secondary forest and shrubs vegetation. *Omphalea trichotoma* (distributed in western and eastern Cuba) and *O. hypoleuca* (reported only from Viñales, Pinar del Río) are endemic (*Greuter & Rankin, 2016*), while *O. diandra* is widely distributed in the neotropics (*Lees & Smith, 1991*). In this archipelago, these plants are present in areas that have been under pressure from the development of infrastructure for tourism and oil extraction activities (*Camacho, Baena & Leyva, 2010*) enhancing the importance of studying their populations and how they may be affected.

Given the direct dependence on host plant availability, *Urania* moths appear to be highly sensitive to changes in land use (*Lees & Smith, 1991*). For instance, *U. sloanus* is now considered extinct due to habitat degradation (*Lees & Smith, 1991*). However, no study has explored how distributional areas of these moths and plants are being affected over time due to land-cover change. This is especially important in the case of the endemic moth *Urania boisduvalii*, due to the risk the host plants are being exposed to. Here, we aimed (1) to evaluate the change in land use in Cuba in the past 35 years, and (2) to analyze how these land use changes may affect the distribution of *Omphalea* plants and *Urania boisduvalii*

moths. Portions of this text were previously published as part of a preprint (*Nuñez-Penichet, Maita-Chamba & Soberón, 2023*; https://www.biorxiv.org/node/3004963.full).

## METHODS

To evaluate land use change in Cuba (Appendix 1 S1), we analyzed the vegetation cover in this archipelago of the years 1985 (beginning of Landsat 5) and 2020 using Landsat images and Google Earth Engine (GEE; *Gorelick et al., 2017*) for the supervised classification processing. GEE, is a cloud geospatial processing platform on Google computational infrastructure, used in different studies with spatial and temporal scales (*Gorelick et al., 2017*). This platform has been used in studies exploring vegetation succession (*Adagbasa & Mukwada, 2022*), species distribution (*Crego, Stabach & Connette, 2022*; *Crego et al., 2023*), and to characterize large landscapes (*Rippel et al., 2023*). GEE has been also used to map deforestation and forest degradation (*Shimizu et al., 2022*; *Wimberly et al., 2022*), predict effects of climate change (*Workie & Debella, 2018*; *Shiff, Lensky & Bonfil, 2021*), assess the impacts of wildfires (*dos Santos et al., 2023*; *Parra et al., 2023*), and detect changes in land use and land cover (*Phan, Kuch & Lehnert, 2020*; *Tassi et al., 2021*; *González-González, Clerici & Quesada, 2022*; *Biswas et al., 2023*).

Our study area (Cuban archipelago) was covered by sixteen scenes (paths: 10 to 17 and rows: 44 to 46) of the Landsat 5 ETM sensor (1985) and the Landsat 8 OLI/TIRS sensor (2020), with images at 30-m resolution. To minimize temporal, spatial, and spectral varying scattering and absorbing effects of atmospheric gasses and aerosols, we used the collection 2 of Land Surface Reflectance (*Vermote et al., 2016*). Cloud-free images of Cuba were produced by using those with cloud cover ≤50% and a temporal filter, including the years 1984–1988 for the period 1985 and the years 2020–2021 for 2020. Cloud and cloud shadows were masked using pixel quality attributes generated from the C Function of Mask (CFMASK) algorithm (QA_PIXEL Bitmask) for both sensors (*Zhu & Woodcock, 2012*; *Foga et al., 2017*).

The resulting images (1985 and 2020) were classified into seven cover types to represent areas in which *Omphalea* populations have been reported (forest and shrubs, considered as suitable) and areas in which *Omphalea* species have not been reported (mangrove, soil without vegetation cover, wetlands, pine forest, agriculture, and water bodies, considered as unsuitable areas for *Omphalea* plants; *Nuñez-Penichet et al., 2016*). The distribution of *Omphalea* in Cuba was estimated by *Nuñez-Penichet et al. (2019)* from post-processing bioclimatic models with suitable cover types and reported as raster files. The cover types were classified by regions of interest (ROI) for each of the categories on the Google Earth Engine. The number of ROIs selected for each category varied to represent the full range of appearances within each category (300 for forest and shrubs, 100 for mangrove, 50 for soil without vegetation cover, 100 for wetlands, 100 for pine forest, 100 for agriculture, and 100 for water bodies). All ROIs were selected manually based on visual interpretation of the respective Landsat images from GEE and were divided randomly into training (70%) and testing (30%).

We selected Random Forest as the classification algorithm as it has been widely used on land use and land cover change investigations (*Alencar et al., 2020*; *Souza et al., 2020*;

*Osman et al., 2023*; *Shimabukuro et al., 2023*). Also, this algorithm is good at handling outliers and noisy datasets, has high processing speed (*Jin et al., 2018*), has good performance with complex datasets (*Belgiu & Drăguţ, 2016*), and has higher accuracy than other algorithms (*Sheykhmousa et al., 2020*; *Talukdar et al., 2020*). The classified images were post-processed in *QGIS.org (2022)* to correct misclassified pixels using a raster calculator and elevational considerations (Appendix 2 S1). We assessed the accuracy of classifications on the GEE platform, using the resulting images from the post-processing analyses and 100 new validation ROIs created randomly for each of the considered cover types but water bodies in which we used only 50. We used measures extracted from confusion matrix, such as: (1) overall accuracy (OA), which is calculated dividing the total number of correctly classified values by the total number of values (*Story & Congalton, 1986*; *Congalton, 1991*), and (2) kappa coefficient (*Fitzgerald & Lees, 1994*; *Fielding & Bell, 1997*) for measuring the agreement between classification and truth values.

The resulting classified images of Cuba were masked to: (1) terrestrial protected areas (*CNAP, 2013*); (2) potential distribution areas of *Omphalea* (from *Nuñez-Penichet et al., 2019*), hereafter OD; and (3) potential distribution areas of the plant within protected areas (*Nuñez-Penichet et al., 2016*). This last one was included as 23.57% of OD is inside a protected area in Cuba. We quantified the percentage of each of the seven cover types and their changes between periods for all Cuba and in each of the masked areas (areas of interest).

To assess the significance of observed changes, we used a bootstrap approach (*Weber & Langille, 2007*) whereby the classification categories were assigned randomly (1,000 times) to both years. With this, we created a distribution of random changes, and compared the observed percentage difference with the distribution of random values. Observed values above the 99 percentiles, or below the first, were considered significantly non-random.

All these analyses were done in *R Core Team (2020)* using the package terra (*Hijmans, 2022*). All the code needed to reproduce these analyses is openly available at https://github.com/claununez/Land-coverChangeCuba. All the data needed is openly available at https://figshare.com/articles/dataset/Data_for_the_manuscript_Land-cover_change_in_Cuba_may_favor_biodiversity_An_example_using_Omphalea_plants_and_Urania_boisduvalii/21779129.

## RESULTS

We obtained an overall accuracy (OA) of 92.7% and 87.6% for 1985 and 2020 classifications, with a Kappa coefficient of 0.91 and 0.85, respectively. The values of OA for each land cover type ranged from 80% (pine forest) to 100% (agriculture) in 1985, and from 71% (wetlands) to 99% (forest and shrubs) in 2020 (Table S1). All the observed values were significant ($p < 0.01$) using the two-tailed bootstrap test described in the methods.

We found that, in the past 35 years, forest and shrub areas have increased across the entire Cuban territory (from 17.02% in 1985 to 38.12% in 2020; Table 1). Agricultural lands and soils without vegetation cover, on the other hand, showed a reduction from 59.62% to 41.26% and from 3.56% to 1.16%, respectively (Table 1 and Fig. 1). The other

**Table 1 Percentage of area classified as one of the seven classification types considered for the years 1985 and 2020.** *Omphalea* includes all the species of *Omphalea* genus distributed in Cuba combined and the areas of potential distribution of *Omphalea* is referring to these species' potential distribution in Cuba. The protected areas in Cuba are referring to the terrestrial protected areas only.

| Classification type | Cuba | | Protected areas in Cuba | | Areas of potential distribution of *Omphalea* | | Areas of potential distribution of *Omphalea* inside protected areas | |
|---|---|---|---|---|---|---|---|---|
| | 1985 | 2020 | 1985 | 2020 | 1985 | 2020 | 1985 | 2020 |
| Forest and shrubs | 17.02 | 38.12 | 29.08 | 37.02 | 47.48 | 70.09 | 59.31 | 72.16 |
| Mangrove | 4.64 | 5.80 | 15.31 | 19.83 | 2.15 | 2.13 | 4.63 | 4.42 |
| Soil without vegetation cover | 3.56 | 1.16 | 2.85 | 0.98 | 1.64 | 0.44 | 1.31 | 0.42 |
| Wetland | 4.08 | 3.68 | 11.39 | 9.85 | 1.40 | 1.25 | 2.03 | 2.07 |
| Pine forest | 1.86 | 1.47 | 3.94 | 2.58 | 6.06 | 3.87 | 11.64 | 7.13 |
| Agriculture | 59.62 | 41.26 | 13.09 | 8.16 | 37.92 | 19.11 | 15.57 | 8.95 |
| Water bodies | 9.23 | 8.50 | 24.34 | 21.58 | 3.35 | 3.11 | 5.53 | 4.85 |

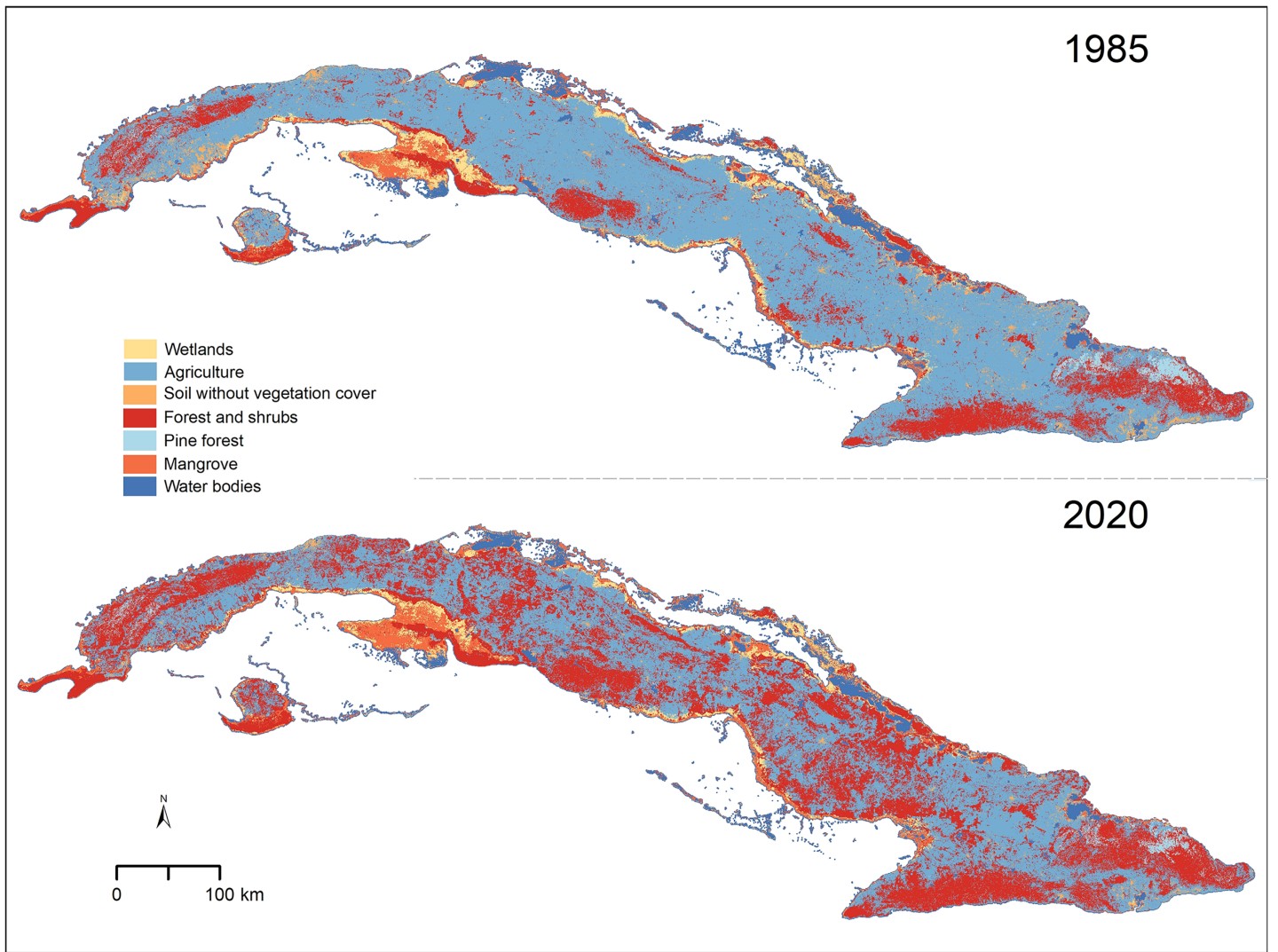

**Figure 1 Vegetation types in Cuba for the years 1985 and 2020.** Cuban silhouette from *ESRI (2016)*.

**Table 2 Percentage of change between the classification types considered, during the period 1985–2020.** *Omphalea* includes all the species of the genus *Omphalea* distributed in Cuba combined and the areas of potential distribution of *Omphalea* is referring to these species' potential distribution in Cuba.

| Type of change | Cuba | Protected areas in Cuba | Areas of potential distribution of *Omphalea* | Areas of potential distribution of *Omphalea* inside protected areas |
|---|---|---|---|---|
| Stable forest and shrubs | 15.01 | 26.40 | 43.72 | 55.25 |
| Mangrove to forest and shrubs | 0.49 | 1.50 | 0.67 | 1.76 |
| Soil without vegetation cover to forest and shrubs | 0.36 | 0.08 | 0.30 | 0.06 |
| Wetlands to forest and shrubs | 0.53 | 0.86 | 0.43 | 0.42 |
| Pine forest to forest and shrubs | 0.94 | 2.02 | 3.18 | 6.06 |
| Agriculture to forest and shrubs | 20.69 | 5.91 | 21.75 | 8.57 |
| Water bodies to forest and shrubs | 0.11 | 0.25 | 0.03 | 0.04 |
| Stable mangrove | 3.67 | 12.23 | 1.27 | 2.46 |
| Forest and shrubs to mangrove | 0.21 | 0.85 | 0.22 | 0.63 |
| Soil without vegetation cover to mangrove | 0.19 | 0.71 | 0.11 | 0.28 |
| Wetlands to mangrove | 0.93 | 3.88 | 0.20 | 0.47 |
| Pine forest to mangrove | 0.00 | 0.00 | 0.00 | 0.00 |
| Agriculture to mangrove | 0.17 | 0.32 | 0.08 | 0.09 |
| Water bodies to mangrove | 0.61 | 1.84 | 0.26 | 0.48 |
| Stable soil without vegetation cover | 0.73 | 0.66 | 0.28 | 0.28 |
| Forest and shrubs to soil without vegetation cover | 0.01 | 0.02 | 0.01 | 0.02 |
| Mangrove to soil without vegetation cover | 0.00 | 0.01 | 0.00 | 0.00 |
| Wetlands to soil without vegetation cover | 0.06 | 0.21 | 0.03 | 0.06 |
| Pine forest to soil without vegetation cover | 0.00 | 0.00 | 0.00 | 0.00 |
| Agriculture to soil without vegetation cover | 0.36 | 0.07 | 0.11 | 0.04 |
| Water bodies to soil without vegetation cover | 0.00 | 0.02 | 0.00 | 0.01 |
| Stable wetlands | 2.36 | 6.12 | 0.65 | 0.95 |
| Forest and shrubs to wetlands | 0.13 | 0.37 | 0.08 | 0.15 |
| Mangrove to wetlands | 0.30 | 1.02 | 0.10 | 0.21 |
| Soil without vegetation cover to wetlands | 0.24 | 0.68 | 0.14 | 0.25 |
| Pine forest to wetlands | 0.00 | 0.00 | 0.00 | 0.00 |
| Agriculture to wetlands | 0.31 | 0.50 | 0.10 | 0.13 |
| Water bodies to wetlands | 0.35 | 1.16 | 0.18 | 0.36 |
| Stable pine forest | 0.74 | 1.80 | 2.50 | 5.23 |
| Forest and shrubs to pine forest | 0.31 | 0.54 | 0.82 | 1.39 |
| Mangrove to pine forest | 0.00 | 0.00 | 0.00 | 0.00 |
| Soil without vegetation cover to pine forest | 0.01 | 0.00 | 0.00 | 0.00 |
| Wetlands to pine forest | 0.00 | 0.00 | 0.00 | 0.00 |
| Agriculture to pine forest | 0.38 | 0.24 | 0.53 | 0.51 |
| Water bodies to pine forest | 0.03 | 0.00 | 0.01 | 0.00 |
| Stable agriculture | 37.50 | 6.04 | 15.23 | 6.22 |
| Forest and shrubs to agriculture | 1.30 | 0.88 | 2.60 | 1.86 |
| Mangrove to agriculture | 0.08 | 0.21 | 0.05 | 0.09 |
| Soil without vegetation cover to agriculture | 1.95 | 0.43 | 0.75 | 0.28 |
| Table 2 (continued) | | | | |
|---|---|---|---|---|
| Type of change | Cuba | Protected areas in Cuba | Areas of potential distribution of *Omphalea* | Areas of potential distribution of *Omphalea* inside protected areas |
| Wetlands to agriculture | 0.18 | 0.23 | 0.08 | 0.10 |
| Pine forest to agriculture | 0.14 | 0.12 | 0.35 | 0.34 |
| Water bodies to agriculture | 0.11 | 0.25 | 0.05 | 0.07 |
| Stable water bodies | 8.00 | 20.83 | 2.82 | 4.56 |
| Forest and shrubs to water bodies | 0.04 | 0.01 | 0.03 | 0.00 |
| Mangrove to water bodies | 0.10 | 0.33 | 0.05 | 0.11 |
| Soil without vegetation cover to water bodies | 0.08 | 0.29 | 0.06 | 0.14 |
| Wetlands to water bodies | 0.03 | 0.10 | 0.01 | 0.03 |
| Pine forest to water bodies | 0.04 | 0.00 | 0.03 | 0.00 |
| Agriculture to water bodies | 0.21 | 0.02 | 0.12 | 0.00 |

four types of cover considered (mangrove, wetland, pine forest, and water bodies) were present in similar proportions in the two scenarios studied (Table 1). Most areas that changed from one cover type to another were located inland, especially in areas of low elevation (Fig. 1 and Appendix 1 S1).

In the areas of potential distribution of *Omphalea* plants, we found a similar pattern to the one described above for Cuba except for the pine forest, which decreased from 1985 to 2020 (Table 1). Most of these changes in the different cover types were concentrated in central Cuba, whereas stable areas were mostly in Guanahacabibes (westernmost part of Cuba), the southern part of Isla de la Juventud, and highlands (Appendix 1 S1 and Fig. S1). Inside the protected areas in Cuba and in the OD that were inside the protected areas, we detected small changes in the percentage of area of each cover type between the two analyzed periods (1985 and 2020) (Table 1, Figs. S2 and S3). The biggest changes were in the forest and shrubs (increased) and in the agricultural lands (decreased) (Table 1, Figs. S2 and S3).

We found that only 15.01% of the Cuban area classified as forest and shrubs in 2020, was classified as the same category in 1985 (Table 2). The low stability of this cover type was also found when quantifying in the other areas of interest, with 26.40% of the protected areas in Cuba, 43.72% of the OD, and 55.25% of the OD inside protected areas found to be forest and shrubs in both time periods analyzed (Table 2). Not many of the forest and shrubs areas changed to other cover types, and most of the forest and shrub areas gained in Cuba in 2020 were agricultural lands in the past (Table 2). The proportion of change from agricultural land to forest and shrubs was also high in the OD and the other areas of interest (Table 2). The agricultural lands that changed to forest and shrubs by 2020 were distributed in small patches across the Cuban archipelago and were less predominant in the eastern region (Fig. 2). The areas that changed to forest and shrubs in OD were also small patches, but their spatial distribution helped increase the connectivity among larger stable forest and shrubs areas (Fig. 2 and Table 2).
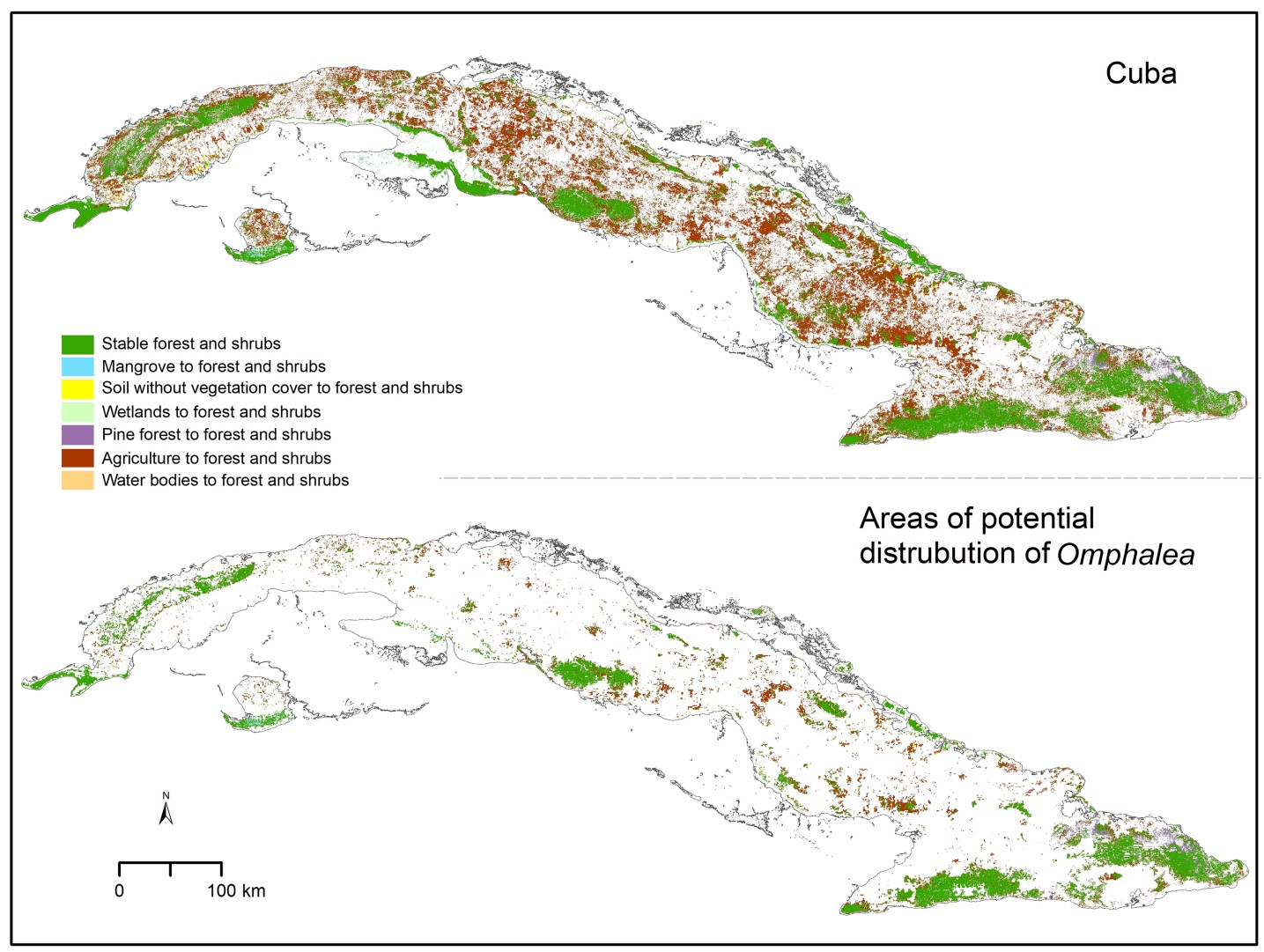

**Figure 2** **Forest and shrubs change.** *Omphalea* includes all the species of *Omphalea* genus distributed in Cuba combined and the areas of potential distribution of *Omphalea* is referring to these species' potential distribution in Cuba. Cuban silhouette from *ESRI (2016)*.

# DISCUSSION

Environmental monitoring sciences have undergone a rapid evolution due to global environmental changes and the technological development of Earth Observation (EO; *Appel et al., 2018*). Several platforms, such as Google Earth Engine (GEE), Sentinel Hub, Open Data Cube, SEPAL, openEO, JEODPP, and pipsCloud, have been developed to address challenges associated with storing, accessing, processing, and analyzing large geospatial datasets generated by EO satellites (*Gomes, Queiroz & Ferreira, 2020*). China, USA, India, and Brazil lead as the countries with higher contributions to the use of GEE for the visualization and processing of geospatial data, while other countries like Cuba, have few or no data showing the use of this platform to address biological questions. Among the EO satellites, Landsat has been the most used, mainly applied to study changes in

cropland-vegetation as well as characterization of land use and land cover (*Velastegui-Montoya et al., 2023*). Landsat images have been previously used in Cuba to study change of cover types (*e.g.*, *Hernández & Cruz, 2016*). Our results show, not only how land cover in Cuba has changed in 35 years, but also how these changes may affect the distribution of *Omphalea* plants and *Urania boisduvalii* moths, using the GEE platform.

The general global trend in forest cover is towards loss (*Keenan et al., 2015*), and cover loss is associated with changes in plant-herbivore networks. However, we found that in Cuba, forest and shrubs (*i.e.*, suitable vegetation cover for *Omphalea*) have increased considerably in the past 35 years, mostly replacing agricultural lands (Fig. 1 and Table 1). Although there have been some reforestation efforts in specific localities of Cuba (*Gebelein, 2012*; *Izquierdo et al., 2015*) that may have influenced the changes in soil without vegetation cover observed, our results may be due to the fact that in Cuba, before 1990, extensive areas were dedicated to growing sugarcane, especially in the center of the island (Appendix 1 S1), but after 1990, the intensity of sugarcane production decreased (*Suárez et al., 2012*; *Machado, 2018*). This explanation was also presented by *Clark, Aide & Riner (2012)* after finding a 10% increase in closed-canopy forest area, mainly in the Cuban Dry Forests ecoregion, from 2001 to 2010, and by *Álvarez-Berríos et al. (2013)* while analyzing changes in Cuba woody areas and agricultural lands. Our results also concur with *Sagastume et al. (2018)* who reported that over half of agricultural land in Cuba is currently unused and with *Stuhlmacher et al. (2020)* who found that between the years 1985 and 2010, ~18% of Cuban croplands changed to barren/grass/shrublands, built-up lands, and forest. Although the trend we report for Cuba is opposite to the general global pattern, it is an example of how land use change is a complex phenomenon resulting from multiple causes, and difficult to predict (*Lambin et al., 2001*).

Forest and shrub cover increase within areas of potential distribution of *Omphalea* (Table 1 and Fig. S1) may in fact represent an advantage for populations of these plants as these areas are known to be climatically suitable (*Nuñez-Penichet et al., 2016*). Given the obligated relationship between *Urania boisduvalii* and *Omphalea* plants, which also drives the moth's dispersal patterns in the country (*Nuñez-Penichet, Soberón & Osorio-Olvera, 2023*), these gained areas could indirectly benefit the moth. The increment of forest and shrubs could increase connectivity among fragmented patches with plant populations facilitating *Urania* moth's movements (Figs. 2 and S1). In fact, several of the stable and gained shrub and forest patches are located in the potential migratory paths of *U. boisduvalii* identified in a previous study (*Nuñez-Penichet et al., 2019*).

On the other hand, the increment of forest and shrubs in the Cuban territory does not necessarily imply an increase of *Omphalea* plants or an increment in favorable conditions for *Urania* moths. Species distributions depend on the presence of suitable abiotic conditions, accessibility, and biotic interactions (*Soberón & Peterson, 2005*). In Cuba, a significant portion of unused agricultural lands have been reported to be covered with shrub invasive species (*e.g.*, *Dichrostachys cinerea*, *Sagastume et al., 2016*, *2018*; *Valero-Jorge et al., 2024* and *Vachellia farnesiana*, *Fernandez et al., 2018*), which causes a displacement of native plant communities (*Ruiz, Remond & Fernandez, 2010*). In the last decades, several studies have focused on testing the potential of *D. cinerea* high-quality

biomass to produce a sustainable biofuel (*Sagastume et al., 2018*; *Reyes et al., 2022*) as an alternative source of energy, and therefore little effort is targeted on controlling this invasive plant. For this reason, performing fieldwork in the gained forest and shrubs areas will be necessary to assess whether they can actually benefit the conservation of *Omphalea* plants and *Urania* moths.

When we focused on the protected areas in Cuba and areas of potential distribution for *Omphalea* inside protected areas, we detected that, in the two time-periods analyzed, stability was high for all cover types explored. The exceptions were the increase of forest and shrubs, and the decrease of agriculture areas and soils without vegetation cover (Figs. S2 and S3). This concurs with what has been previously reported for 10 National Parks in Cuba (*Hernández & Cruz, 2016*). In fact, these results may reflect the role of the protective areas in preserving the landscape in Cuba. However, interpretations about the conservation status of natural vegetation should be done cautiously. In our study, and based on our goal, we included all types of forests and shrubs in a single category, and therefore, we are not accounting for changes at a finer detail (*e.g.*, changes from tropical rain forest to secondary vegetation dominated by alien species). Because of this, forest and shrub gains do not necessarily represent positive rates of change for primary or native vegetation.

Here we are presenting results that suggest the distribution of *Omphalea* plants and, therefore, for *Urania* moths in Cuba may be increasing, since we find that the areas with suitable conditions for these species have been increasing considerably. Moreover, our results suggest that land-use changes in Cuba may favor not only *Urania* and *Omphalea* but other species that live in forests and shrubs. However, the presence of invasive species in those gained forest and shrubs areas should be monitored in the field as they may represent a serious threat to native biodiversity.

Changes in land use in Cuba during the last 35 years may represent a significant opportunity for biodiversity conservation and sustainable management of its natural vegetation. However, further studies that allow understanding the community composition and structure in such forests and shrubs are needed to assess the value of these lands for biodiversity conservation in the country.

## ACKNOWLEDGEMENTS

We thank Marlon E. Cobos for his useful comments on the manuscript.

### Funding

This work was supported by the project "Restauración y Dinámica de los Ecosistemas Andino-Amazónicos del Sur del Ecuador" (08-DI-FARNR-2021). The funders had no role in study design, data collection and analysis, decision to publish, or preparation of the manuscript.

## Grant Disclosures

The following grant information was disclosed by the authors:

Restauración y Dinámica de los Ecosistemas Andino-Amazónicos del Sur del Ecuador: 08-DI-FARNR-2021.

## Competing Interests

The authors declare that they have no competing interests.

## Author Contributions

- Claudia Nuñez-Penichet conceived and designed the experiments, performed the experiments, analyzed the data, prepared figures and/or tables, authored or reviewed drafts of the article, and approved the final draft.
- Juan Maita conceived and designed the experiments, performed the experiments, analyzed the data, authored or reviewed drafts of the article, and approved the final draft.
- Jorge Soberon conceived and designed the experiments, authored or reviewed drafts of the article, and approved the final draft.

## Data Availability

All the code needed to reproduce these analyses is available at GitHub and Zenodo:

- https://github.com/claununez/Land-coverChangeCuba.

- Claudia Nuñez-Penichet. (2024). claununez/Land-coverChangeCuba: Land-cover change in Cuba (v1.0.0). Zenodo. https://doi.org/10.5281/zenodo.11252550.

All the data needed is available at figshare: Nuñez-Penichet, Claudia; Soberon, Jorge (2022). Data for the manuscript: Land-cover change in Cuba and how it is affecting the areas of distribution of Omphalea (Angiosperma: Euphorbiaceae) and *Urania boisduvalii* (Lepidoptera: Uraniidae). figshare. Dataset. https://doi.org/10.6084/m9.figshare.21779129.v2.

## Supplemental Information

Supplemental information for this article can be found online at http://dx.doi.org/10.7717/peerj.17563#supplemental-information.

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
