# Peer review of "Land-cover change in Cuba and implications for the area of distribution of a specialist’s host-plant"

_PeerJ, doi:10.7717/peerj.17563_

## Round 0.1 · original submission · Major Revisions

· Academic Editor

Major Revisions

Dear Dr. Nuñez-Penichet,

After this first review round, the reviewers believe the manuscript has merits to be published. Still, there is room for improvement. Although the general English is good, at least two reviewers indicated a grammatical review is necessary. Also, there is some need to improve the introduction section as well.

One of the authors also indicated that a clearer history covering the interspecific interaction involving the plant would be interesting to see. Also, more information on how the distribution for Omphalea was determined is needed as well. Still, there were also more improvements to be made.

Please do not forget to resubmit your manuscript along with a rebuttal letter informing the editor and the reviewers about the changes that were implemented and those that were not. Please do not forget to justify the cases when the suggestions were not implemented.

Sincerely,
Daniel Silva

Reviewer 1 ·

Basic reporting

The manuscript is well structured and written in clear and easily understandable language. However, the reader would profit from a grammar check of the text, as e.g. often commas are missing (e.g. line 20 (after "Here"), line 76 (after "boisduvalii") or line 146 (after "areas")). Besides, when literature is cited within the main text, the bracket should only include the year, e.g. line 118: "from Nuñez-Penichet et al. (2016)" instead of "from (Nuñez-Penichet et al., 2016)".
Some further suggestions to improve readability:
• You differentiate in your classification between “forest and shrubs” and “pine forest” but there are no details about the land use classes given. For a better understanding of the ecological context, it might help to describe which type of plants make up the non-pine forest and how this forest community differs from pine forests
• line 28: "has" instead of "have"
• line 43: There is a lot of more recent literature on this topic. I would choose some papers published in the last couple of years to underline the topicality of this issue
• line 54-57: I assume from your description that plants only start producing metabolites upon feeding, so that they are not toxic from the beginning. Maybe state this more explicitly, because right now it sounds a bit confusing that they would feed on toxic plants.
• line 58: divide into two sentences after the citation of Smith
• line 59-62: is this sentence relevant for the paper? Maybe remove it, you cannot say anything about structural complexity from 30m images and it does not seem to play a role in any further part of the manuscript.
• Line 82: “in” instead of “of”
• Line 94: maybe remove “and”
• Line 98: It might sound better to say instead: “to represent the full range of appearances within each category”
• Line 104: “to have” or “having”
• Line 106: maybe “and because of its high…”? for better readability
• Line 117: “of” instead of “for”
• Line 131: there is a square in the middle of the word, I guess it should be “coefficient”
• Line 150: 5.01% is probably supposed to be 15.01%
• Line 188: There is a bracket missing, either behind 2018 or at the end of the phrase
• Line 191: I would rather say “little effort is targeted on controlling”
• Figure 1: The color scheme is not very intuitively, it might make it easier to grasp the different land use types by changing colors, e.g., forest in green
• S1 Appendix 1: It might help to get an impression of the study area by adding mean annual temperature and precipitation as not every reader might be familiar which climatic conditions are implied by a neotropical location

Experimental design

The study is well implemented, based on a profound image analysis, and addresses with land use change one of the main current issues of global change. The observed changes in overall land use in Cuba which contrast the trends in most other parts of the world is a very interesting aspect, well supported by the data and clearly the strength of the analysis. This is also the focus when looking at the figures but from the title and introduction, I would rather expect analyses concerning changes within the plant and moth distribution areas. As the analyses only consider potential distribution areas of the host plant genus and do not include any direct distribution data about the moth, the data is a bit thin to draw reliable conclusions about effects on the host plants or the moth. Therefore, I think the manuscript would profit from adjusting the story line by focusing more clearly on the overall land use changes in Cuba and then introducing the plant-moth interaction as an example for how land cover changes might be a suitable tool to assess effects on specialist species or other aspects of biodiversity. In this context, I suggest to adjust the title to something like “land-cover change in Cuba and implications for the area of distribution of a specialist’s host-plant”, as from “how it is affecting the areas of distribution” I would expect changes in the distribution areas of Omphalea, caused by land use change, which is not analyzed in the manuscript.

Further, it does not become clear to the reader why you do a separate comparison of land cover change within protected areas. This should be apparent from the introduction and be discussed in the discussion. You only mention the comparison for the overlapping areas between protected areas and the Omphalea distribution in the discussion. However, also for this comparison I am lacking a conclusion what can be drawn from it. It is an interesting aspect that land use in general changed less within protected areas and might be a good point to be added to the main story line, discussing the temporal stability of land cover within protected areas in comparison to the strongly changing covers in the rest of the country. It potentially indicates a positive effect of protected areas e.g., for species like Omphalea, as they guarantee stable patches of forest occurrence in the landscape, which support those species (according to your discussion Line 183) and might act as stepping stones to distribute into the newly forest-covered areas. Also, you mention in the results that there was a low temporal stability of forest and shrub category, which is the suitable habitat area in your case, but in the discussion, you do not address this issue. Further, in this context, it might be interesting to know the area or percentage of Omphalea distribution area that falls within protected areas.

Validity of the findings

All underlying data and scripts have been provided and are clearly and easily understandably presented.

·

Basic reporting

There is no issue with the English used throughout the manuscript. The general format also seems fundamentally sound. All raw data is provided publicly.

The article is well-referenced with regards to general ecology, the ecology of Cuban ecosystems, and in the potential ecological impacts of understanding recovery after changes in land-use policies.

However, the article appears to be missing a literature review of the body of work leading to the analyses used in the study. That is, there has been much research into developing geospatial methods for analyzing large areas of land, particularly the use of Google Earth Engine. For examples, see articles such as the following: (https://doi.org/10.3390/rs12152411), (https://doi.org/10.3390/rs13122299), (https://doi.org/10.1007/s12237-023-01192-z), etc.

It would be nice to see some background literature review on geospatial analyses to ensure that the article has well-rounded science underlying all of the methods and results.

Experimental design

The analyses presented in the article are relevant to the field and ecological applications. The authors set out to answer a simple question that has not been answered before for Cuba and their methods are rigorous and technical enough to provide a succinct answer. Thus, the current article methods are potentially in a state to be published without much additional material.

However, I was curious when reading the article and looking through the materials if there were additional data available for other periods of time that the authors could look into to provide more robust context to recovery after land-use changes. It would be interesting to potentially see some sort of time-series analysis looking at multiple dates of imagery, particularly if the imagery from Landsat is available for the region at different points in time.

With a time-series analysis for land class changes, the authors would be able to provide more detail on how potential habitat areas for Uraniidae moths have changed over time and also be able to provide some speculation as to how these areas may change in the future.

Validity of the findings

The findings for this article are valid for the level of analysis performed and the rigor needed to answer their the authors' research goals. As stated above, some inclusion of additional points in time may lend to a deeper understanding of land-use trends over time. However, the questions seem adequately answered as the reporting stands for this submission, especially considering this study is a first look into land recovery that is prudent to direct future research.

The article provides no form of statistical testing for the authors' land classifications and their quantification of potential habitats for Urania boisduvalii. The authors could potentially research a bit into the drivers of land-use recovery to quantify drivers for positive changes in potential habitat. As it stands, the study does not provide validation of the data they have generated other than classification accuracy or confidence that these data are the result of underlying trends in the ecology or usage of land by entities in Cuba. This assumes that classifications are direct representations of land classes present in the field without considering whether there is any ecological significance related to them.

Furthermore, is the assumption that habitat "recovery" to potential habitat is associated with true community or species-level recovery? This may be a strong assumption to make without collecting supporting data on the presence of Uraniidae moths. It has often been found in ecological surveys that the recovery of land and the reintroduction of usable host species does not result in recovery of the populations which used those habitats in the past. It would be good to include some speculation or literature at the end of the discussion for potential findings of future work so that readers are more aware of a potential disconnect between trophic levels of these ecosystems.

Additional comments

The same disconnect between the reporting and the geospatial methods used by the authors occurs in the discussion. The reader should get an understanding, from the literature review of the introduction and comparison to other work in the discussion, of how the work of this article relates to other geospatial modeling classification papers. It should be clearer that this study builds on the body of work present in the field that has been performed in other areas using similar methods. If this article adds to those methods, it should also be stated in relation to the other literature. Many readers are not going to understand the importance of using Google Earth Engine or the studies that have pioneered its use across geospatial literature.

Also, why is your method appropriate for understanding the land-use changes of your system? Be absolutely clear on this.

Finally, is there any potential relationship with climate changes for the land changes in this system? It would be beneficial to make some note of this potential as many land changes across the world are dictated by shifts in climate such as in precipitation or seasonal temperature variation.

·

Basic reporting

Overall, I found the language throughout to be relatively clear. However, there were some instances where clarity could be improved:
L42: What do the authors mean by “landscape degradation”? This is a very vague term and the authors do not define it. Throughout the manuscript, I was confused as to whether this refers to habitat loss (like is implied on L67-69), or something else (e.g. invasive plants, L185), and I often felt like they were conflated.
L56-59: “[Omphalea] produce secondary metabolites as a defense mechanism against Urania herbivory (Lees & Smith, 1991). These secondary compounds make the plants toxic for the moths, forcing them to move to a different host-plant population (Smith, 1983), therefore, Urania needs to have several patches of Omphalea available to guarantee its survival.” <- I believe the authors need to be more clear here that the production of metabolites happens AFTER herbivory has begun (at least, that’s what I believe happens); otherwise it is confusing to the reader how the moths feed at all.
L96-102: The region of interest explanation is unclear; why did the authors need to select them and what were they used for? Were they used as training data for Google Earth to classify? Or only for accuracy assessments?
L146: “we detected more changes in the percentage of area of each cover type between the two analyzed periods” <- more changes relative to what? Wording is unclear.
L150: “We found that only 5.01% of Cuba classified as forest and shrubs in 2020, was classified under the same category in 1985 (Table 2).” <- should this be 15.01%? Also, this wording is a little unclear; do you mean only XX% of the pixels classified as forest and shrubs in 2020 were classified as that same category in 1985?
L178: “This is because the stability and increment of [forest and shrub]” <- what do you mean by stability and increment? Also, doesn’t calling it stable contradict L150 where the authors argue that only 15% of current forest pixels existed in 1985 and write about “The low stability of this cover type”? Also, I think the authors need to point out that not much forest and shrub area was converted to any other land type through time in their Results section.
L198-202: This is very vague, and I cannot discern what the authors are trying to argue here. Could they be more explicit or reword to clarify?
I also found that the introduction was too brief to give enough context to the study. The authors could provide some more context on land cover loss; what are overall trends? It would also benefit from some background and references on plant-animal interactions in the context of habitat loss. What is the state of the literature? Most importantly, it would benefit from some necessary natural history information on Omphalea/Urania. What kind of plant is Omphalea? What specific habitat requirements does it have? What is the basis for the evidence that Urania feeds exclusively on Omphalea? What is the range size of Omphalea plants? Is there indication that the ranges are declining? Is Omphalea sensitive to land cover change? Why is it expected to decline in response to land cover change?

Experimental design

I think this manuscript would greatly benefit from more details as to how the area of distribution for Omphalea was originally determined; it is completely obscure as it stands. Specifically:
- Are the potential distribution areas pixels or polygons? What spatial resolution are they?
- How did original authors figure out potential distribution for Omphalea? My concern is that it is circular: that potential distribution was determined by clipping to a specific habitat/land cover type, and then this manuscript is measuring change in that land cover type and making inferences about how that habitat change is affecting Omphalea.
- Agriculture is classified as ‘unsuitable’ for Omphalea but the authors, and yet almost 20% of the potential distribution of Omphalea is within agriculture.
- When was the area of distribution of Omphalea calculated? Do you only have 1 value? If it used to be more widely distributed than it is in the AOD that you used, you could be underestimating how much land use has affected the genus. Also, because you only have 2 time points for land cover change, if much of the land cover change happened early on (i.e. before the distributions were calculated), how would this affect your results?

Validity of the findings

My main concern with this paper is whether much of the conclusions are well-supported by the analysis. There were a few instances where I felt the wording was misleading about what the authors actually did/found. The authors measured land cover change WITHIN potential areas of distribution of Omphalea, whereas the wording was ambiguous enough that it sounded like they had measured distributional changes in the plant/moth themselves. For example:
Title: “Land-cover change in Cuba and how it is affecting the areas of distribution of Omphalea (Angiosperma: Euphorbiaceae) and Urania boisduvalii
L77-78: “We… analyze how these land use changes may affect the distribution of Omphalea plants and Urania boisduvalii moths.”
L203: “Here we are presenting results that suggest the distribution of Omphalea plants and for Urania moths in Cuba may be increasing”
Abstract: “We found that the category of forest and shrubs have increased significantly in Cuba in the past 35 years” <- the word “significant” is misleading as you haven’t measured statistical significance.
Secondly, there are no statistics performed on the findings, which makes it hard to tell if some of these changes are significant. For example, did any of the land cover types increase/decrease in a statistically significantly manner? Was the proportional gain in forest and shrub area similar within Omphalea potential distribution vs. Cuba as a whole, and is this statistically significant? Was the proportional change from agriculture -> forest and shrubs statistically similar/different in all 4 areas of interest? Was land cover within protected areas statistically more stable on average than outside?
Thirdly, the authors discuss that the increased forest area means increased connectivity (ex. L160, L178) but don’t actually quantify it. I think the manuscript would benefit from a connectivity analysis, to bolster the relevance to Omphalea/Urbania.

Additional comments

L82: Why begin in 1985? Presumably that is because that is when the satellite imagery begins, but the authors need to provide justification.
L93-96: Why is ‘pine forest’ separate from the ‘forest and shrub’ category? You don’t explain to the reader why some of these categories are unsuitable vs. suitable for Omphalea.
What is the proportion of Omphalea’s range that is currently in a protected area?
L182-194: This paragraph starts off by saying an increase in forest area is not necessarily imply an increase in suitable habitat for Urbania/Omphalea because unused agricultural lands tend to have invasive species. But wouldn’t a decrease in agricultural lands (as the authors found) be beneficial then for reducing the presence of invasive species? Or can the invasive species the authors mentioned persist in forest/shrubs as well?
L196: Why is there low stability for soils without vegetation cover?
L198: “However, interpretations about the conservation status of natural vegetation inside protected areas should be done cautiously, as given the goal of our study, we included all types of forests and shrubs in a single category.” <- shouldn’t interpretations about the conservation status of natural vegetation be interpreted cautiously OUTSIDE of protected areas, for the same reason?

---

## Round 0.2 · Minor Revisions

· Academic Editor

Minor Revisions

Dear Dr. Nuñez-Penichet,

Your manuscript has been well-received by the two reviewers at this time. Still, minor reviews are needed in the introduction, results, and discussion that prevent the acceptance of our manuscript at this time. Please take the due diligence to correct the issues raised by Reviewer #3 before the manuscript is accepted for publication.

Sincerely,
Daniel Silva

·

Basic reporting

The authors have adequately addressed comments from the reviewers about language and literature cited.

Experimental design

The authors have adequately addressed or responded to my concerns about the methodology.

Validity of the findings

The discussion now covers many concerns the reviewers had about the interpretation of the authors' results.

Additional comments

I do not have any major concerns about the acceptance of this manuscript.

·

Basic reporting

Please note that the line numbers do not match up between the authors’ response and the manuscript, which made it very difficult to compare what the authors said they did in the response to the edited document.

I am still unsatisfied with the level of detail provided by the authors on how the how the area of distribution for Omphalea was originally determined. I don’t think it’s enough to put one sentence about it. A reader should not have to read another paper to understand what the authors did in this paper.

In my original comments, I noted that I found that “the introduction was too brief to give enough context to the study. The authors could provide some more context on land cover loss; what are overall trends? It would also benefit from some background and references on plant-animal interactions in the context of habitat loss. What is the state of the literature?” The authors’ response was that they added a line to their discussion; this line simply states that forest loss is associated with “changes” in plant-herbivore networks, which is very vague.

Note that the abstract mentions that “forests and shrubs” are increasing, but does not mention that that is considered suitable habitat, so it’s unclear from the abstract alone how the increase will affect the plant/moth. It seems completely unconnected as it stands.

Experimental design

I appreciate the authors adding a statistical analysis (bootstrap) and I believe this bolsters their findings.

Validity of the findings

I believe the revised title is much more suitable.

---

## Round 0.3 · accepted · Accept

· Academic Editor

Accept

Dear Dr. Nuñez-Penichet,

I am pleased to accept your manuscript for publication in PeerJ!

Congratulations.

Sincerely,
Daniel Silva